# The Influence of Parenting Styles on Eating Behavior and Caries in Their Children: A Cross-Sectional Study

**DOI:** 10.3390/children9060911

**Published:** 2022-06-17

**Authors:** María José González-Olmo, Ana Ruiz-Guillén, María Moya-López, Martín Romero-Maroto, María Carrillo-Díaz

**Affiliations:** 1Department of Orthodontics, Rey Juan Carlos University, 28922 Alcorcón, Spain; mariajose.gonzalez@urjc.es (M.J.G.-O.); martin.romero@urjc.es (M.R.-M.); 2Department of Paediatric Dentistry, Rey Juan Carlos University, 28922 Alcorcón, Spain; maria.moya@urjc.es (M.M.-L.); maria.carrillo@urjc.es (M.C.-D.)

**Keywords:** parenting style, anxiety, eating behavior, caries

## Abstract

The type of parenting style influences the way children cope with problems and can create states of anxiety that can even alter their eating habits, which can cause problems in their oral health. The present study aimed to find out which parenting style is more favorable for the oral health of their children. In this cross-sectional study, 186 children (aged 8–15 years) were examined to assess the mean Decayed/Missing/Filled teeth (DMFT + dmft) index, and they were asked to complete the State–Trait Anxiety Inventory, the Parenting Style Scale, and questions about their oral hygiene habits. On the other hand, their parents answered the Parental Feeding Style Questionnaire and the Children’s Eating Behavior Questionnaire. Results showed that a worse oral health status was associated with a higher state of anxiety, more overeating, more emotional eating, and higher psychological control. A higher rate of missing teeth was associated with increased undereating and overeating. Emotional eating was positively related to psychological control. State of anxiety, overeating, and parental psychological control predicted 24.6% of DMFT + dmft. In addition, emotional eating had a moderating effect in DMFT + dfmt only in those with low levels of affection and communication. In conclusion, high psychological control and low levels of parental affection and communication will increase the state of anxiety in children, influencing their caries rate.

## 1. Introduction

Children’s oral health is influenced by intrinsic, family, and environmental factors. Due to the close interaction between the child and the family, when an oral health diagnosis is made in children, there should be a comprehensive evaluation not only of caries and concomitant clinical situations, but also of factors at the family level [1].

Baumrind [2] distinguishes between three possible types of parenting, on which numerous subsequent studies have been based: authoritarian, authoritative, and permissive. The authoritarian parenting style is characterized by very high levels of demand, where the opinions of children are not taken into account, and children have no option to express themselves; there is no communication between parents and children. The authoritative parenting style is the one that has the greatest positive impact on children, as well as on the relationships they establish with their parents. These parents have a close, respectful, and empathetic relationship with their children, with a high degree of control and with great signs of affection and bidirectional communication. Permissive parents are characterized by indulgence and great displays of affection, but rules and limits are absent [3,4].

This classification of parenting styles has served as a useful tool for investigating the influence of parenting on various issues related to child development, including anxiety [5]. Anxiety is defined as a negative affective state resulting from the perception of a threat, characterized by a perceived inability to predict, control, or obtain desired outcomes in upcoming situations; it is related to stress and can alter the course or outcome of chronic diseases [6].

Since mental health conditions affect the whole body including oral health, it is thought that anxiety may influence the incidence of oral diseases [7]. Some behavioral changes associated with anxiety, such as poor oral hygiene or increased sugar consumption, have been shown to lead to plaque accumulation and the development of periodontal disease, as well as increasing the risk of other dental problems such as caries [6]. Given that sugar consumption is increasing among children, influencing children’s oral and general health [3], it is important to know what children’s and parents’ behaviors may be responsible. 

There are coping mechanisms, such as dietary responses, that regulate and reduce the negative emotions produced by anxiety, which can influence oral health. These mechanisms may be inadequate coping, such as “emotional eating” (EE), or a loss of control over food [8]; in emotional eating, it is not hunger, time, or social needs that cause people to eat but emotions [9]. Several children’s eating behaviors have been related to the incidence of caries; among them are “emotional undereating”, which reflects the decrease in eating response as a result of various negative feelings, such as sadness, anger, or anxiety [10], and “emotional overeating”, where individuals lose control over their eating, and they start to binge, as eating provides distraction and solace from painful negative emotions [8]. In addition, it has been shown that the type of parenting can directly influence eating outcomes [11].

Therefore, since there is evidence to support a potential relationship between parenting styles, child eating behavior, and dental caries [4], and since research on this topic to date is very limited, the aim of the present study was to find the relationship between the type of parenting and the eating behavior disorders of children and to examine whether this had consequences for the caries rate of the children.

## 2. Materials and Methods

### 2.1. Study Design and Setting

This observational, descriptive, and cross-sectional design research was carried out in Spain from January to March 2022. The study sample consisted of 186 subjects, with an age range of 8 to 15 years and their parents. The subjects were selected randomly, and the sample size was calculated based on the results of the prevalence of caries in this region [12], a prevalence of caries at 12 years of age in the permanent dentition of 30.1%. A random sample selection of 172 subjects was estimated to be representative of the cohort at that age, with a 95% confidence level and based on 3 degrees of precision. The value of 172 individuals would be optimal if it were a randomized study; however, as it was a convenience sample, this value was taken as a reference.

Data were collected on children and adolescents attending the Dental Clinic of the Universidad Rey Juan Carlos for dental checkups. The clinic offers a free oral examination to the population without access restrictions and with affordable fees that allow a high percentage of the population to have access to oral treatment. As the patients were waiting their turn in the waiting room, a member of the research team explained the philosophy and purpose of the study and asked them if they wanted to participate anonymously, voluntarily, and without additional compensation. If the mother/parent/guardian agreed, an informed consent form was given, and the instructions of the questionnaire were explained. In addition, a researcher was available to answer questions about the survey. The exclusion criteria established for this study were as follows: participants with local or systemic diseases that could affect oral health; for example, physical or psychological disabilities, chronic medication intake, medication with a high sugar content or that altered salivary flow and/or composition, patients with dental appliances, participants with defects in the structure of the enamel and dentin, participants whose mother tongue was not Spanish, and those who did not provide signed informed consent. On the other hand, the inclusion criteria defined were all participants aged between 8 and 15 years who came to the dental clinic for a check-up and who did not present any of the exclusion criteria mentioned above.

Before starting, three examiners were trained and calibrated by a specialist. The result of the kappa statistics during the calibration process was 0.91 (good agreement).

All subjects gave their informed consent for inclusion before they participated in the study. The study was conducted in accordance with the Declaration of Helsinki, the protocol was approved by the Ethics Committee of Rey Juan Carlos University (2004202111821), and informed consent was obtained from the patients’ parents.

### 2.2. Measures

In the present study, basic sociodemographic aspects (sex, age, and socioeconomic level), hygiene habits, emotional eating behavior data, state of anxiety, and parenting style were collected by means of a structured self-administered questionnaire; in addition, an oral examination was performed on all the participants to record the DMFT index (sum of decayed teeth (DT), missing teeth (MT), and filled teeth (FT)) [13]. As the patients had mixed dentition, the DMFT and dmft indices were summed, as previously seen in other studies [14].

Anxiety symptomatology of children was evaluated as a trait using the state of anxiety subscale of the State–Trait Anxiety Inventory (STAI) [15]. Participants reported their anxiety levels on a 0-point Likert-type scale, where 1 meant “rarely” and 3 meant “almost always”. The state of anxiety subscale consists of 20 items. A total score was calculated by summing the responses of each child, with a range of 0–60 (higher scores indicate more anxiety). Cronbach’s alpha reliability was 0.86.

The Parenting Style Scale [16] was used to evaluate the educational style of the parents. Specifically, two subscales were used: the parental affection and communication subscale and the psychological control subscale. Children were asked to indicate which response best defined their relationship with their parents; the response format was a 6-point Likert-type scale, with a response range from 1 (strongly disagree) to 6 (strongly agree). The sum of the child’s responses (8–48) for each subscale was calculated to determine a final score on the questionnaire, with higher scores being indicative of greater affection and communication or greater psychological control. Cronbach’s alpha reliability was 0.79 for the affection and communication subscale and 0.80 for the psychological control subscale.

The emotional eating (EE) variable was recorded through three subscales of two different questionnaires: the Parental Feeding Style and the Children’s Eating Behavior Questionnaires. Parents completed these questionnaires.

Parental Feeding Style Questionnaire (PFSQ): Five items corresponding to the emotional eating dimension of the Parental Feeding Style Questionnaire were used [17]. These 5 items assessed the degree to which the children had been emotionally nurtured by the parents. The response format was a 5-point Likert-type scale, with a response range from 1 (never) to 5 (always). The sum of parental responses (5–25) was calculated to determine a final score on the questionnaire, with higher scores being indicative of greater emotional nurturance provided by parents. Cronbach’s alpha reliability was 0.77 for this scale.

The Children’s Eating Behavior Questionnaire (CEBQ) was used [18], in particular, we used the version validated in Spanish [19]. Two subscales of this questionnaire were used to assess whether emotional overeating or undereating occurred. Each consisted of 4 items, the response format was a 5-point Likert-type scale, with a response range from 0 (never) to 4 (always), with a range from 0 to 16. A higher score on the emotional overeating subscale indicated a greater tendency to increase intake in negative emotional contexts. The same was true for the emotional undereating subscale; a higher score indicated a greater tendency to reduce intake in negative emotional contexts. Cronbach’s alpha reliability was 0.79 for the overeating subscale and 0.82 for the undereating subscale.

The two questionnaires assessing EE were translated into Spanish, and an additional back translation was carried out to ensure fidelity to the original items. Children’s understanding of the items was previously tested by conducting interviews with subjects of similar age to those in the sample.

Hygiene habits. Children were asked. “In general, how often do you brush your teeth?” The response format was a 5-point Likert-type scale, ranging from 1 (I do not usually brush my teeth) to 5 (three times a day).

Use of dental floss. The response format was dichotomous (yes/no).

Regular visits to the dentist. Children were asked: “In general, how often do you go to the dentist?”. The response format was a 5-point Likert-type scale, (1 = I have never gone; 2 = only when I have a problem or pain; 3 = every 2 or 3 years; 4 = once a year, and 5 = every 6 months.

Oral health status. Pediatric dentists from the University Clinic conducted clinical examinations and determined the oral health status of each participant using a flat-surface oral mirror, scanning probe, and compressed air. In addition, the World Health Organization (WHO) index for the number of decayed, missing, and filled in permanent and deciduous dentition (DMFT + dmft) was calculated [13,14]. The decayed/missing/filled teeth index is one of the simplest and most used indices in epidemiologic surveys of dental caries. It quantifies dental health status based on the number of carious, missing, and filled teeth.

### 2.3. Statistical Analysis

This descriptive study considers the variables described in the previous section. Statistical analysis was performed using SPSS v26 (SPSS Inc., Chicago, IL, USA). The data analysis included descriptive statistics and the Kolmogorov–Smirnov test to evaluate the assumption of normality, which was confirmed. To understand the possible differences, *t*-tests were performed. Scheffé (if equal variance was assumed) and Games–Howell (if not) post hoc tests were used, and effect sizes were calculated. For the t of independent samples, a Cohen’s d was performed. According to Cohen (1988), small Cohen’s d values are ≈0.2, medium ones are ≈0.5, and high ones are ≈0.8. Cohen [20] also considers small effect size values to be ≈0.01, medium ones to be ≈0.06, and those large enough to be taken into account as ≈0.14. The relationships between variables were analyzed using Pearson’s correlations. A hierarchical linear regression model was performed to determine the predictors of DMFT + dmft. Subsequently, the Hayes PROCESS module (version 3.3) was used to perform multiple simple moderation analyses (Model 1) [21]. Cronbach’s alpha was also performed to evaluate the internal consistency of the instruments. The significance was set at *p* < 0.01.

## 3. Results

### 3.1. Sociodemographic Characteristics

The sample was composed of 186 subjects (101 girls and 85 boys), with an average age of 12.8 ± 1.46 years, DMFT + dmft of 3.9 ± 3.9, an average number of teeth with caries of 2 ± 2.9, an average number of filled teeth of 1.7 ± 2.3, and an average number of teeth extracted of 0.1 ± 0.4. Of the sample, 10.3% belonged to the low socioeconomic level, 87% to the medium, and 2.7% to the high.

Regarding daily dental hygiene, 68.9% of the sample brushed their teeth more than twice a day, while only 22% used dental floss, and 79.1% attended dental checkups at least once a year. All patients used fluoride toothpaste of 1450 ppm.

The mean and standard deviation of the state of anxiety, subscale of undereating, subscale of parental emotional eating, and the subscale of parental affection and communication and psychological control are described in Table 1.

Significant differences between the sexes were found in the subscale of overeating (higher in girls) and affection and communication (higher in girls). Differences were found for missing teeth (higher in children). No significant differences were found for the rest of the variables (Table 1). Moderate/large effect sizes were observed in all comparisons.

### 3.2. Relationship among the Variables of Anxiety State, Overeating, Undereating, Emotional Eating, Affection and Communication, Psychological Control, and Oral Health

As can be seen in Table 2, a higher DMFT + dmft was associated with a higher state of anxiety, more overeating, more EE, higher psychological control, less frequent visits to the dentist, and lower socioeconomic level. A higher rate of missing teeth was associated with increased undereating and overeating. A higher rate of decayed teeth was positively associated with higher psychological control, poorer oral hygiene, less frequent visits to the dentist, and lower socioeconomic level. Overeating was positively related to a higher state of anxiety, more undereating more EE, poorer oral hygiene, and less frequent visits to the dentist. With more affection and communication, the less psychological control, and the better oral hygiene, EE was negatively related to affection and communication and positively related to psychological control and lower socioeconomic level.

### 3.3. Predictor Variables of DMFT + dmft

Hierarchical multiple regression was performed to determine whether the sum of the state of anxiety, overeating, and parental psychological control significantly predicted DMFT + dmft. Model 1 (state anxiety) gives an R^2^ of 0.134. Model 2 (overeating) gives an increased R^2^ of 0.064. The full model of the state of anxiety, overeating, and parental psychological control (Model 3) was statistically significant with R^2^ = 0.259, F (1, 184) = 21.156, *p* < 0.001; adjusted R^2^ = 0.246. State of anxiety, overeating, and parental psychological control predicted 24.6% of DMFT + dmft. See Table 3 for complete details on each regression model.

### 3.4. Moderation Analysis of Affection and Communication on EE and DMFT + dmft

A moderation analysis was performed with the EE levels as the independent variable, the DMFT + dmft as the dependent variable, and affection and communication as the moderating variable. The regression analysis, in which the EE levels were considered as predictors of DMFT + dmft, was obtained (b = 0.18, s.e. = 0.1, *p* = 0.07), and it was the same for the moderator variable (b = 0.01, s.e. = 0.07, *p* = 0.93); however, a significant value was obtained for the interaction between the independent and moderator variable (b = −0.05, s.e. = 0.02; (−0.09, −0.01), *p* < 0.01;). Obtaining a significant value for this interaction indicates the presence of a moderating effect, suggesting that affection and communication interfere with the effect of EE on DMFT + dmft.

To determine when affection and communication had a moderating effect on the main model, the level of significance and the upper and lower limits were analyzed. In this case, we observed that EE had a moderating effect only in those with low levels of affection and communication (*p* < 0.01; s.e. = 0.12; (0.15, 0.61)). However, it had no effect on those with medium levels (*p* = 0.2; s.e. = 0.1; (−0.07, 0.33)) nor for high levels (*p* = 0.87; s.e. = 0.13; (−0.28, 0.23)) (Figure 1). In summary, we observed that affection and communication interfered with the effect of EE on the DMFT + dmft index.

## 4. Discussion

This study examined the various parental behaviors, children’s eating behaviors, and their relationship to the incidence of dental caries. The results of the present study are consistent with past findings that the EE and the oral health of children are significantly affected by parenting style. Parenting style has longitudinal effects on the behavior of the child [22]. As observed in other studies [23], positive parenting styles, such as authoritative parenting, had a positive effect on children; this style includes bidirectional communication, high parental affection, allowing children some autonomy, and using positive discipline instead of punishment [24]. On the contrary, in previous studies, and coinciding with the results obtained in the present study, poorer communication and less affection, as well as greater psychological control by parents (authoritarian parenting style), were associated with an adverse impact on children’s oral health [3,23,25]. 

The level of discipline has been found to have a negative correlation with the prevalence of caries; on the contrary, an overly democratic parenting style leads to self-confidence in the child, which is not conducive to better oral health [25]. Therefore, an authoritative parenting style has been related to better oral health of children and lower prevalence of caries, due to a bidirectional communication [1,24]. Authoritarian parents have high demands that they place on their children, but this does not imply high levels of responsiveness (less communication) [1,24]. Results from previous studies [3,26] show that children of authoritative parents have a lower caries rate compared to permissive and authoritarian parents, indicating that parental behavior may have an influence on the development of children’s oral habits. Although some studies have shown that the permissive parenting style depicted a threefold increase in caries status when compared to the authoritative parenting style [24], there are also studies that found no association between parental style and caries rate [27]. 

According to the study conducted by Sahithya and Raman [5], the authoritative parenting style decreases the likelihood of children having anxiety; this may be due to a greater understanding and more stable behavior on the part of the child; therefore, authoritarian styles (high psychological control and less affect) and permissive (coddling children) will be risk factors for the appearance of anxiety in children. In addition, parental style is an essential determinant of the child’s coping style [3]. Anxiety states and coping problems have been shown in previous studies [10] to influence children’s feeding response, also affecting the prevalence of caries. When children’s coping does not develop correctly, negative coping mechanisms, such as emotional eating, may appear. Goodman et al. [11] found that authoritative parenting styles will be a protective factor against negative eating behaviors. A higher psychological control was associated with higher emotional eating [28]. However, there are also studies that did not find an association between parenting style and emotional eating [29].

The results obtained in the present study show that EE (emotional overeating and emotional undereating) led to a higher caries rate, coinciding with the results of previous studies [10], which also showed that there was a relationship between greater parental control and an increase in the caries rate, coinciding with the findings of this study. In addition, children with greater emotional and behavioral difficulties had a higher prevalence of tooth decay and being overweight [30]. Regarding sex, several studies have shown [8,31] that it is easier for girls to develop EE, coinciding with the results obtained in the present study. 

The main limitation of the present study is that this study has a cross-sectional design, which cannot support causality and mediating effects. In addition, although many variables are included, it excludes many environmental factors, in addition to the plaque levels of the patients. Therefore, the results should be interpreted with caution, since caries is a multifactorial chronic disease, and other risk factors may influence it in addition to parental style. Another possible limitation is the use of self-reported measures, which may be affected by recall and comprehension bias: and responses based on social desirability. Finally, a convenience sample was used, which was obtained from a specific segment of the population in the community of Madrid, potentially limiting the possibility of generalizing the results.

Future lines of research are required to utilize longitudinal studies to assess whether parenting styles influence children’s emotional response to food, and whether the rate of tooth decay is associated with parenting.

## 5. Conclusions

In conclusion, according to the results obtained, high psychological control and low levels of parental affection and communication will increase the state of anxiety in children, which may provoke the need to create coping mechanisms, such as EE, which can increase the caries rates. A negative correlation was found between EE and affection and communication with children, and a positive correlation was found between EE and psychological control; likewise, higher levels of EE were found to influence a worse oral health status.

Thus, it can be concluded that understanding the role of parental style as a risk factor for developing poor oral health habits will help health professionals to develop preventive strategies that motivate parents with different behavioral styles. In addition, it is of great importance to know how different parenting styles influence children in order to provide adequate treatment and effective behavioral management of pediatric patients.

## Figures and Tables

**Figure 1 children-09-00911-f001:**
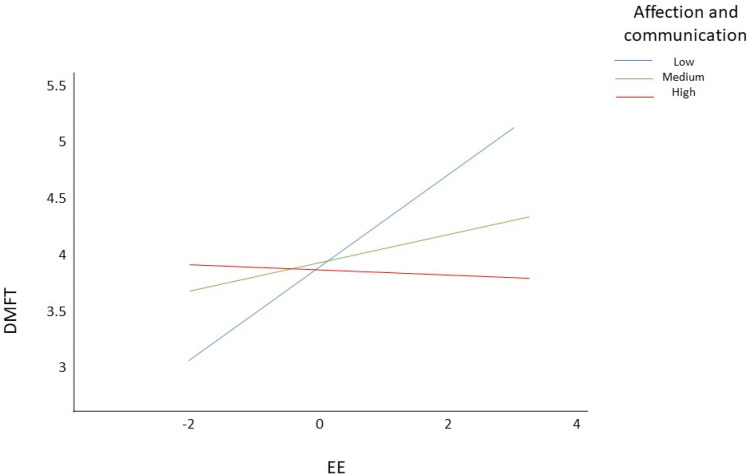
Analysis of moderation of affection and communication on EE and DMFT.

**Table 1 children-09-00911-t001:** Mean and standard deviation in decayed, missing, filled, DMFT + dmft, anxiety state, overeating, undereating, emotional eating, affection and communication, and promotion of autonomy. Comparative by gender.

	Mean (DS)	Boys Mean (DS)	Girls Mean (DS)	t (*p*)	D Cohen
Anxiety state	6.1 (5.8)	5.7 (4.9)	6.4 (6.4)	0.8	0.12
Overeating	2.3 (3.1)	1.9 (2.8)	3.4 (3.2)	3.3 **	0.49
Undereating	5 (3.7)	4.7 (3.9)	5.4 (3.4)	1.2	0.19
Emotional eating	7 (2.9)	7.1 (3.1)	7 (2.8)	0.1	0.03
Affection and communication	43.9(4.1)	43.2 (3.7)	44.5 (4.2)	1.2 *	0.32
Psychological control	21.7 (8.7)	22.1 (7.8)	21.3 (9.4)	0.5	0.09
Decayed	2 (2.8)	2.1 (2.7)	2 (2.9)	0.03	0.03
Missing	0.1 (0.4)	0.2 (0.5)	0.1 (0.1)	3 **	0.27
Filled	1.7 (2.3)	1.7 (2.5)	1.8 (2.1)	0.2	0.04
DMFT + dmft	3.9 (3.9)	4.1 (4.1)	3.9 (3.7)	0.1	0.05

Note: * Correlation is significant at the 0.05 level. ** Correlation is significant at the 0.01 level.

**Table 2 children-09-00911-t002:** Pearson’s Correlation between decayed, missing, filled, DMFT + dmft, anxiety state, overeating, undereating, emotional eating, affection and communication, promotion of autonomy, hygiene habits, use of dental floss, regular visits to dentist, and socioeconomic level.

	α	1	2	3	4	5	6	7	8	9	10	11	12	13	14
Anxiety state	0.86	1	0.254 ** 0.000	−0.059 0.427	0.070 0.341	−0.165 * 0.025	0.105 0.154	0.411 ** 0.000	0.096 0.191	0.079 0.285	0.367 ** 0.000	−0.217 ** 0.000	−0.064 0.387	−0.141 0.055	−0.090 0.227
Overeating	0.79		1	0.316 ** 0.000	0.215 ** 0.003	−0.113 0.125	0.101 0.168	0.370 ** 0.000	0.170 * 0.020	0.083 0.261	0.337 ** 0.000	−0.252 ** 0.000	0.014 0.847	−0.195 ** 0.008	−0.112 0.130
Undereating	0.82			1	0.259 ** 0.000	−0.106 0.148	0.218 0.003	0.027 0.715	0.285 ** 0.000	0.072 0.327	0.101 0.168	0.129 0.084	−0.157 * 0.032	0.153 * 0.031	0.184 * 0.131
Emotional eating	0.77				1	−0.151 * 0.040	0.163 * 0.026	0.22 ** 0.002	0.14 0.052	−0.039 0.598	0.160 * 0.029	−0.141 0.061	−0.034 0.641	−0.086 0.242	−0.150 * 0.042
Affection and communication	0.79					1	−0.292 ** 0.000	−0.044 0.548	0.073 0.324	−0.075 0.311	−0.075 0.309	0.260 ** 0.000	−0.040 0.59	−0.019 0.79	−0.002 0.98
Psychological control	0.76						1	0.333 ** 0.000	0.032 0.660	0.077 0.294	0.302 ** 0.000	−0.090 0.219	0.094 0.204	−0.316 ** 0.000	−0.062 0.405
Decayed								1	0.280 ** 0.000	0.032 0.669	0.791 ** 00.000	−0.194 ** 0.008	−0.066 0.370	−0.394 ** 0.000	−0.289 ** 0.000
Missing									1	0.046 0.535	0.344 ** 0.000	−0.048 0.514	−0.143 0.051	−0.099 0.180	−0.088 0.236
Filled										1	0.625 ** 0.000	0.033 0.652	0.203 * 0.005	0.135 0.065	−0.031 0.674
DMFT + dmft											1	−0.125 0.089	0.046 0.531	−0.226 ** 0.002	−0.244 ** 0.001
Hygiene habits												1	0.044 0.549	0.318 ** 0.000	0.206 ** 0.005
Use of dental floss													1	−0.019 0.794	0.004 0.952
Regular visits to the dentist														1	0.153 * 0.038
Socioeconomic level															1

Note: * Correlation is significant at the 0.05 level. ** Correlation is significant at the 0.01 level.

**Table 3 children-09-00911-t003:** Results of regression analysis to predict DMFT + dmft from state anxiety, overeating, and parental psychological control.

	DMFT + dmft					
	Model 1		Model 2		Model 3	
Variable	B	β	B	β	B	β
Constant	2.4 **		1.8		−0.4	
State anxiety	0.2 **	0.3	0.2 **	0.3	0.1 **	0.2
Overeating			0.3 **	0.2	0.2 **	0.2
Psychological control					0.1 **	0.2
R^2^	0.134		0.198		0.259	
F	28.5 **		22.5 **		21.1 **	
∆R^2^	0.134		0.064		0.061	
∆F	28.5 **		14.5 **		14.8 **	

Note. *N* = 184, ** *p* < 0.01.

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
