# Peer review of "The Influence of Parenting Styles on Eating Behavior and Caries in Their Children: A Cross-Sectional Study"

_children, 2022, doi:10.3390/children9060911_

Round 1

Reviewer 1 Report

Line 21 - you mention DMFT value of 24.6% - "State of anxiety, overeating, and parental psycho-21 logical control predicted 24.6% of DMFT." The value cannot be reconciled as it is with none of the results. If such mentioned, it should have been presented similar within the result area. 

Line 94 - please clarify the inclusion/exclusion criteria for the subjects analyzed within the study.

Line 108 - please clarify / describe accurate the DMFT determination process, for both mixed and permanent dentition, as your subjects belong to 8-15 age range.

Line 175, Line 214, Line 217 - as you introduce the concepts for Model 1 and Model 3, Model 2 should also be introduced before being used in the Table 3.

Line 295 - according to the aim of the study, line 71-72, "aim of the present study was to find the relationship between the type of parenting and the eating behavior disorders of children and to examine whether this had consequences for the caries rate of the children". The aim (and also title) is partially sustained by the conclusions - please review and improve.

Author Response

Line 21 - you mention DMFT value of 24.6% - "State of anxiety, overeating, and
parental psychological control predicted 24.6% of DMFT." The value cannot be
reconciled as it is with none of the results. If such mentioned, it should have been presented similar within the result area.

- Dear reviewer, thank you for your appreciation. The value of 24.6 % DMFT
has been clarified in the Results section behind the adjusted R2 = .246. Line
233-234.

Line 94 - please clarify the inclusion/exclusion criteria for the subjects analyzed
within the study.

- Following the reviewer's suggestions, the exclusion and inclusion criteria have
been clarified in the Material and Methods section.

Line 108 - please clarify / describe accurate the DMFT determination process, for
both mixed and permanent dentition, as your subjects belong to 8-15 age range.

- We agree with the reviewer's suggestion, which is why we have proceeded to
clarify that the patients had mixed dentition (lines 117-118, lines169-170), and
that for this reason, and based on previous studies that have carried out this
method, we have proceeded to carry out a sum of the DMFT and the dmft.

Line 175, Line 214, Line 217 - as you introduce the concepts for Model 1 and
Model 3, Model 2 should also be introduced before being used in the Table 3.

- Details on model 2 and model 3 have been added in the Results section. Line
228-229.

Line 295 - according to the aim of the study, line 71-72, "aim of the present study was to find the relationship between the type of parenting and the eating behavior disorders of children and to examine whether this had consequences for the caries rate of the children". The aim (and also title) is partially sustained by the conclusions - please review and improve.

- Thanks for the suggestion. The conclusions have been specified in more detail.

Reviewer 2 Report

Parental styling and eating habits have been well studied in the literature.

In this paper, with a convenient sample (that means that the results cannot be generalized to the population which is a limitation of the study), the authors ask questions regarding hygiene habits, the use of dental floss and regular visits to the dentist as well as socioeconomic level. It is very well known that all these variables could affect the DMFT of the children but they were not analyzed in the study and were not taken in consideration in the statistical analysis. I suggest to analyze these variables too and to study the correlation.

I addition, the answer to the question to the child "how many times do you brush your teeth"? (could be bias since the child will try to answer what is expected not always the true) it is very interesting that "Regarding daily dental hygiene, 68.9% of the sample brushed their teeth more than  twice a day". Did the authors checked level of plaque? This can give a better idea of the teeth brushing. In addition, did the children brushed the teeth with a fluoridated paste 1450 ppm or above?

The question: Regular visits to the dentist. Children were asked: "In general, how often do you go to the dentist?”. The response format was a 5-point Likert-type scale, with a range from 1 (have never gone) to 5 (every 6 months). It is not clear what is 2, 3 and 4? In addition is it reliable that the child answers every 6 months? Does an 8 years old child really know how often does he visits the dentist?

Author Response

Parental styling and eating habits have been well studied in the literature.

In this paper, with a convenient sample (that means that the results cannot be generalized to the population which is a limitation of the study), the authors ask questions regarding hygiene habits, the use of dental floss and regular visits to the dentist as well as socioeconomic level. It is very well known that all these variables could affect the DMFT of the children but they were not analyzed in the study and were not taken in consideration in the statistical analysis. I suggest to analyze these variables too and to study the correlation.

  • The variables of hygiene habits, flossing and regular visits to the dentist, as well as socio-economic status have been added to the correlation analysis. It has been added to Table 2 and to the Results section. Line 213-221.

In addition, the answer to the question to the child "how many times do you brush your teeth"? (could be bias since the child will try to answer what is expected not always the true) it is very interesting that "Regarding daily dental hygiene, 68.9% of the sample brushed their teeth more than  twice a day". Did the authors checked level of plaque? This can give a better idea of the teeth brushing. In addition, did the children brushed the teeth with a fluoridated paste 1450 ppm or above?

  • Thank you for your suggestions. Indeed, the questionnaires present a limitation which is the response due to social desirability that had been explained in the limitations. As for the examination of the plaque level, it was not carried out in the study, so it has been decided to include it as a limitation of the study. Finally, regarding the question of fluoride toothpaste, the question was specifically:

"Do you use fluoride toothpaste of 1450 ppm or higher?" With a dichotomous yes/no response.

This question was not described in the material and methods section since all participants answered yes, so we thought it would not contribute anything interesting to the correlations. However, this has been specified in the Results section.

The question: Regular visits to the dentist. Children were asked: "In general, how often do you go to the dentist?”. The response format was a 5-point Likert-type scale, with a range from 1 (have never gone) to 5 (every 6 months). It is not clear what is 2, 3 and 4? In addition is it reliable that the child answers every 6 months? Does an 8 years old child really know how often does he visits the dentist?

  • Following the reviewer's suggestions, we have made the clarifications of the range of response in the manuscript being 1=I have never gone, 2=only when I have a problem or pain, 3=every 2 or 3 years, 4=once a year and 5=every 6 months. In addition, despite the fact that a member of the research team was in the waiting room in case the participant had any doubts and their mother/father supervised them, it was decided to include this data in the Limitations section as a comprehension

    Parental styling and eating habits have been well studied in the literature.

    In this paper, with a convenient sample (that means that the results cannot be generalized to the population which is a limitation of the study), the authors ask questions regarding hygiene habits, the use of dental floss and regular visits to the dentist as well as socioeconomic level. It is very well known that all these variables could affect the DMFT of the children but they were not analyzed in the study and were not taken in consideration in the statistical analysis. I suggest to analyze these variables too and to study the correlation.

    • The variables of hygiene habits, flossing and regular visits to the dentist, as well as socio-economic status have been added to the correlation analysis. It has been added to Table 2 and to the Results section. Line 213-221.

    In addition, the answer to the question to the child "how many times do you brush your teeth"? (could be bias since the child will try to answer what is expected not always the true) it is very interesting that "Regarding daily dental hygiene, 68.9% of the sample brushed their teeth more than  twice a day". Did the authors checked level of plaque? This can give a better idea of the teeth brushing. In addition, did the children brushed the teeth with a fluoridated paste 1450 ppm or above?

    • Thank you for your suggestions. Indeed, the questionnaires present a limitation which is the response due to social desirability that had been explained in the limitations. As for the examination of the plaque level, it was not carried out in the study, so it has been decided to include it as a limitation of the study. Finally, regarding the question of fluoride toothpaste, the question was specifically:

    "Do you use fluoride toothpaste of 1450 ppm or higher?" With a dichotomous yes/no response.

    This question was not described in the material and methods section since all participants answered yes, so we thought it would not contribute anything interesting to the correlations. However, this has been specified in the Results section.

    The question: Regular visits to the dentist. Children were asked: "In general, how often do you go to the dentist?”. The response format was a 5-point Likert-type scale, with a range from 1 (have never gone) to 5 (every 6 months). It is not clear what is 2, 3 and 4? In addition is it reliable that the child answers every 6 months? Does an 8 years old child really know how often does he visits the dentist?

    • Following the reviewer's suggestions, we have made the clarifications of the range of response in the manuscript being 1=I have never gone, 2=only when I have a problem or pain, 3=every 2 or 3 years, 4=once a year and 5=every 6 months. In addition, despite the fact that a member of the research team was in the waiting room in case the participant had any doubts and their mother/father supervised them, it was decided to include this data in the Limitations section as a comprehension

This manuscript is a resubmission of an earlier submission. The following is a list of the peer review reports and author responses from that submission.